# Scalable synthesis enabling multilevel bio-evaluations of natural products for discovery of lead compounds

Lizhi Zhu[1,2], Wenjing Ma[1], Mengxun Zhang[1], Magnolia Muk-Lan Lee[2], Wing-Yan Wong[2], Brandon Dow Chan [2], Qianqian Yang[1], Wing-Tak Wong[2,3,4], William Chi-Shing Tai[2,3] & Chi-Sing Lee [1]

Challenges in the development of anti-cancer chemotherapeutics continue to exist, particularly with respect to adverse effects and development of resistance, underlining the need for novel drugs with good safety profiles. Natural products have proven to be a fertile ground for exploitation, and development of anti-cancer drugs from structurally complex natural products holds promise. Unfortunately, this approach is often hindered by low isolation yields and limited information from preliminary cell-based assays. Here we report a concise and scalable synthesis of a series of low-abundance *Isodon* diterpenoids (a large class of natural products with over 1000 members isolated from the herbs of genus *Isodon*, which are well-known folk medicines for the treatment of inflammation and cancer), including eriocalyxin B, neolaxiflorin L and xerophilusin I. These scalable syntheses enable multilevel bio-evaluation of the natural products, in which we identify neolaxiflorin L as a promising anti-cancer drug candidate.

[1] Laboratory of Chemical Genomics, School of Chemical Biology and Biotechnology, Peking University Shenzhen Graduate School, Shenzhen University Town, Xili, Shenzhen 518055, China. [2] Department of Applied Biology and Chemical Technology, Hong Kong Polytechnic University, Hung Hom, Hong Kong, China. [3] State Key Laboratory of Chinese Medicine and Molecular Pharmacology (Incubation), Shenzhen Research Institute of The Hong Kong Polytechnic University, Shenzhen 518057, China. [4] State Key Laboratory of Chirosciences, Department of Applied Biology and Chemical Technology, Hong Kong Polytechnic University, Hung Hom, Hong Kong, China. Correspondence and requests for materials should be addressed to W.-T.W. (email: w.t.wong@polyu.edu.hk) or to W.-S.T. (email: william-cs.tai@polyu.edu.hk) or to C.-S.L. (email: lizc@pkusz.edu.cn)

Cancer is the second leading cause of death worldwide, accounting for 8.8 million deaths in 2015 according to the World Health Organization (WHO). In 2012, there were 14 million new cases of cancer[1], and this number is only set to rise, with a predicted 70% increase in incidence over the next twenty years. Chemotherapy is one of the major modalities for cancer treatment and natural products have proven their worth in this area, serving as springboards for drugs[2] such as dactinomycin, doxorubicin, vincristine and Taxol (paclitaxel). Despite their success in extending patient survival, the presence of adverse effects and development of resistance hinder their therapeutic values[3–5]. Identifying leads from natural products for further development has proven challenging as many are low-abundance and showing similar levels of in vitro anti-cancer activity in preliminary cell-based assays. In the search for novel lead compounds in the midst of low-abundance natural products, scalable synthesis that enables multilevel bio-evaluation is highly desirable[6].

*Isodon* diterpenoids (Fig. 1) are a large class of natural products with over 1000 members isolated from the herbs of genus *Isodon* (well-known folk medicines for the treatment of inflammation, pneumonia, cancer and also respiratory and gastrointestinal disorders) and have been seen as a promising source of leads for anti-cancer therapeutics[7,8]. Oridonin is one of the most-studied *Isodon* diterpenoids for cancer treatment due to its abundance, but its moderate potency and bioavailability slowed down its development as an anti-cancer therapeutic[9,10]. Recently, eriocalyxin B (**1**, an oridonin variant) has been reported to exhibit potent anti-cancer effects, and is a promising candidate for further preclinical development[11]. Oridonin and eriocalyxin B (**1**) are both 7,20-epoxy-*ent*-kauranoids, which contain the tetracyclic core of *ent*-kaurene with a C7-C20 hemiketal bridge, forcing the

boat conformations of the B and C rings (Fig. 1). This conformation allows intramolecular hydrogen bond formation between the C6 β-hydroxyl and the C15 carbonyl, which is important for the anti-cancer activity according to the structure–activity relationship studies[7,12]. The intriguing structure and biological activity of *Isodon* diterpenoids has attracted considerable efforts towards their synthesis[13–52]. However, only two total syntheses of 7,20-epoxy-*ent*-kauranoids have been published including the pioneering work reported by Mander's group on (±)-15-desoxy longikaurin C[49,50] and by Reisman's group on (−)-longikaurin E[51,52]. Because of this, we have decided to develop a dependable synthesis towards 7,20-epoxy-*ent*-kauranoids, especially those with low isolation yields[53], for a detail study of their anti-cancer activities.

Here we report a concise and scalable total synthesis of a series of low-abundance 7,20-epoxy-*ent*-kauranoids including (±)-eriocalyxin B (**1**), (±)-neolaxiflorin L (**2**) and (±)-xerophilusin I (**3**) that enables multilevel bio-evaluation of this subclass of *Isodon* diterpenoids. Although (±)-neolaxiflorin L (**2**) exhibits only moderate cell growth inhibitory activity, it is identified as a promising lead candidate for further anti-cancer drug development due to its remarkable efficacy in animal study with no apparent toxicity.

## Results

**Strategy**. Our synthetic strategy towards *Isodon* 7,20-epoxy-*ent*-kauranoids involved construction of the tetracyclic core rapidly via an iterative ene-type cyclization strategy. As shown in Fig. 2a, the alkene of **I** could undergo Diels-Alder (DA) cycloaddition with **II** forming the decalin (AB ring system) of **III** with an alkene regenerated in the B ring. The CD ring system is

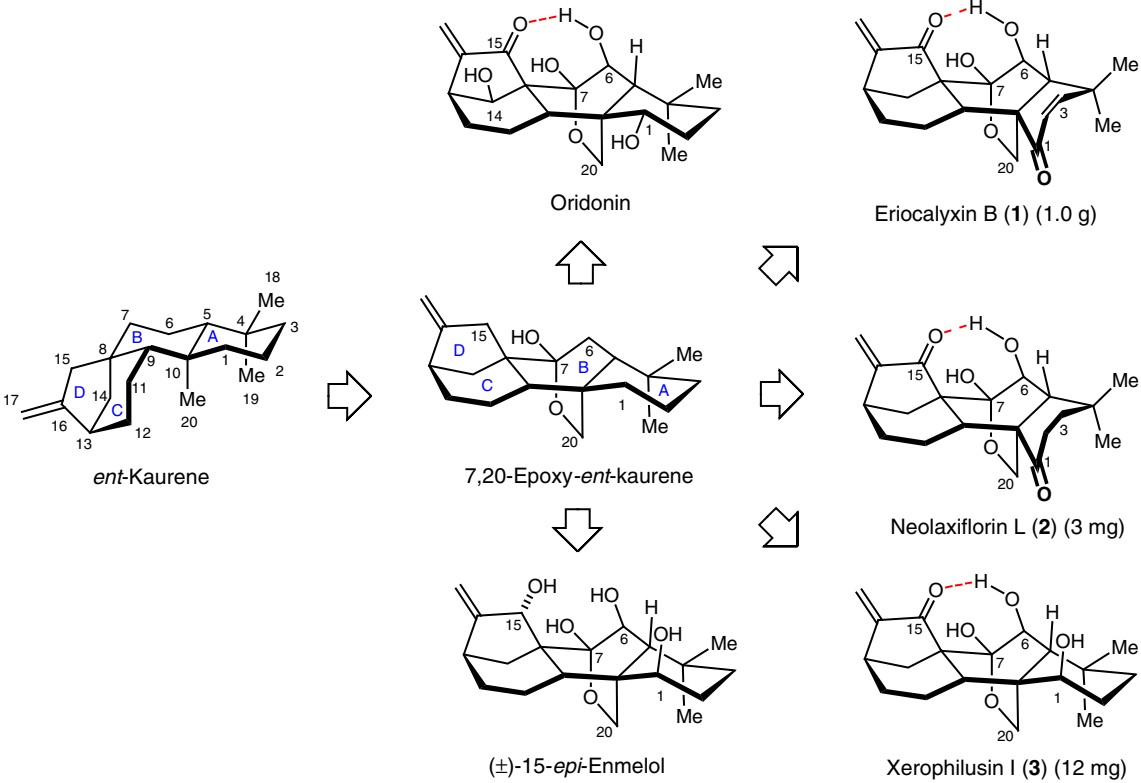

**Fig. 1** Representative *Isodon* 7,20-epoxy-*ent*-kauranoids. This subclass of *Isodon* diterpenoids features a C7-C20 hemiketal bridge, boat conformations of the B and C rings, and intramolecular hydrogen bonding between the C6 β-hydroxyl and the C15 carbonyl. The numbers in parenthesis indicate the amount that isolated from 10 kg of dried leaves of *l. eriocalyx* var. *laxiflora*[53]

**Fig. 2** Synthetic strategies. **a** The tetracyclic core of *Isodon* 7,20-epoxy-*ent*-kauranoids could be established via an iterative ene-cyclization strategy. **b** The appropriate oxygenation could be installed via a site-selective oxidation and reduction sequence via a conformation and hydrogen-bond-guided redox–relay strategy

anticipated to be established via intramolecular Mukaiyama–Michael of **III** followed by carbocyclization of **IV** in a cascade manner. Through this sequence of ene-type cyclizations, the alkene regenerated in the C ring of **IV** and then ended up as an exeocyclic alkene in the D ring of **V**. After establishing the tetracyclic core of **V**, the appropriate oxygenation pattern at C1, C6, C12 and C15 could be installed selectively via a conformation and hydrogen-bond-guided redox–relay strategy (Fig. 2b)[54]. After a preliminary conformational analysis, the less hindered C12 ketone of **V** is expected to reduce selectively. The residual C7 ketone could allow α-oxygenation for installation of the C6-ketone, which is important for equilibration of the *cis*-AB ring system to the *trans*-AB ring system. The C7 hydroxyl of **VI** could direct the reduction at C6 and the allylic C-H bond oxidation at C15 via the effects of hydrogen bonds. Finally, the C1 hydroxyl of **VII** is anticipated to oxidise selectively due to the hydrogen bonding between the C6 hydroxyl with the C15 ketone.

**Total Syntheses of *Isodon* 7,20-epoxy-*ent*-kauranoids 1–3.** The synthesis began with pyridinium chlorochromate (PCC) oxidation of (±)−**4** follow by olefination with phosphonate **5** using 1,8-diazabicyclo(5.4.0)undec-7-ene (DBU) and NaI (Fig. 3). This reaction initially gave a *E:Z* mixtures (2.5:1) of enone (±)−**6** with 67% yield, which can be converted quantitatively to the *E*-isomer upon treatment of HCl in aqueous tetrahydrofuran (THF). Conversion of enone (±)−**6** to the corresponding silyl enol ether followed by DA cycloaddition with **7** in refluxing toluene afforded DA product (±)−**8** as a mixture of diastereomers (*exo:endo* = 5:1). Reduction of (±)−**8** facilitated the separation process and

afforded diol (±)−**9** with 58% yield (2 steps in one-pot from (±)−**6**). After protection of the diol as *t*-butyldimethyl (TBS) ethers, treatment of dichlorodicyanoquinone (DDQ) led to enone (±)−**10** in a single operation. Unfortunately, Mukaiyama–Michael/carbocyclization cascade cyclization attempts of (±)−**10** using dual-mode Lewis acids[55,56] (such as $Zn^{2+}$, $Fe^{3+}$ and $In^{3+}$) did not give any of the expected cyclized product (±)−**11**. After a survey of a variety of strong σ-Lewis acids, we found that only $Me_2AlCl$ in $CH_2Cl_2$ afford trace amounts of (±)−**11** along with the Michael adduct as the major side-product. To our delight, we finally found that using the combination of $Me_2AlCl$ and LiBr can greatly enhance the subsequent carbocyclization step and afford 65% of (±)−**11** in a single operation. In our previous study, using $Me_2AlCl$ alone cannot induced carbocyclization of silyl enol ethers with alkynes[55]. There is only one successful example reported in the literature by using $EtAlCl_2$ as the promoter, which gave the 6-*endo* cyclization product[57]. The mechanism of this highly efficient cascade cyclization is not clear and a detail mechanistic study of this transformation is ongoing in our laboratory. The stereochemistry of (±)−**11** was determined by comparison with the X-ray structure of (±)−**12**, which was obtained upon treatment with *p*-toluenesulfonic acid (TsOH) in dichloromethane. This iterative ene-type cyclization strategy required only seven steps to establish the tetracyclic core of (±)−**11** with 20% overall yield in decagram scales from readily available substrates (±)−**4**, **5** and **7**, which were prepared in one-pot from commercially available materials (decagram scales).

According to the conformational analysis, the C12 carbonyl of (±)−**11** is less hindered, which was selectively reduced and

**Fig. 3** Synthesis of the tetracyclic core. The tetracyclic core of (±)−**11** was established via an iterative ene-type cyclization strategy from readily available substrates (±)−**4**, **5** and **7** in seven steps with 20% overall yield. The synthetic route can be operated in decagram scales

mesylated to give (±)−**13** (Fig. 4). The C7 carbonyl was then converted to the silyl enol ether, and the mesylate was reduced using LiBHEt₃. Upon treatment of MeLi[58], the enolate generated in situ reacted with oxygen in air and formed the peroxide intermediate, which was reduced by thiourea and provided (±)−**14** with good yields. Interestingly, the nuclear magnetic resonance (NMR) spectral data revealed that the chair conformation of A–C rings in (±)−**13** were equilibrated to the boat conformations of that in (±)−**14**. After oxidation of the resulting C6 hydroxyl to the ketone, the TBS ethers were removed using tetra(*n*-butyl)ammonium fluoride (TBAF). Under this condition, the *cis*-AB ring was equilibrated to the *trans*-AB ring forming the C7,20 hemiketal bridge of (±)−**15**. This four-step sequence efficiently converted the *ent*-kaurene core of (±)−**11** to the 7,20-epoxy-*ent*-kauranoid skeleton of (±)−**15** via conformation guided reduction at C7 followed by oxidation at C6 with 40% overall yield in gram to decagram scales.

With (±)−**15** in hand, allylic C-H bond oxidation at C15 was achieved using SeO₂ with *t*-butyl hydroperoxide (TBHP)[59] providing (±)−**16** in 89% yield as a single diastereomer. The high diastereoselectivity of this reaction could be rationalised by the hydrogen bond between the C7 hydroxyl and the oxidant. As revealed by the X-ray structure of (±)−**16**, the C7-hydroxyl could activate the C6 carbonyl via hydrogen bonding and block the top face of the molecule. Indeed, lithium aluminium hydride (LAH) reduction of the C6 carbonyl gave the β-hydroxyl isomer selectively and afforded (±)-15-*epi*-enmelol (**17**)[60] in good yields. Surprisingly, oxidation of the allylic C15 hydroxyl resulted in a

mixture of unidentified side-products. After screening of various oxidants and conditions, we gratefully found that the C15 allylic hydroxyl can be selectively oxidised using 2-iodoxybenzoic acid (IBX) in a 1:1 mixture of dimethyl sulfoxide (DMSO) and THF, and provided (±)-xerophilusin I (**3**) in 80% yield. Jones oxidation selectively oxidised the C1 hydroxyl[61] and afforded (±)-neolaxiflorin L (**2**) in 92% yield. The C6 hydroxyl was intact due to its hydrogen bond with the C15 carbonyl. This four-step sequence utilised the effects of hydrogen bonds and installed the appropriate oxygenation at C1, C6 and C15 of (±)−**2** selectively from (±)−**15** in 60% overall yield. Finally, Saegusa oxidation[62] of (±)−**2** completed the total synthesis of (±)-eriocalyxin B (**1**). Starting from the key intermediate (±)−**11**, this conformation and hydrogen-bond-guided redox–relay strategy required only 7–10 steps to establish the appropriate oxygenation pattern and afforded (±)−**1**–**3** (in hundred-milligram-to-decagram scales with 19.1–26.3 overall yields).

**In vitro studies**. The anti-cancer activity of natural (−)−**1** and synthetic (±)−**1**–**3** against was evaluated by 3-(4,5-Dimethyl-thiazol-2-yl)-2,5-diphenyl tetrazolium bromide (MTT) assay in a panel of five human cancer cell lines including promyelocytic leukaemia (HL60), hepatocellular carcinoma (SMMC7721), human alveolar basal epithelial adenocarcinoma (A549), breast adenocarcinoma (MCF7) and colon adenocarcinoma (SW480). All four *Isodon* diterpenoids exerted significant growth inhibition against all cell lines tested, in a dose dependent manner

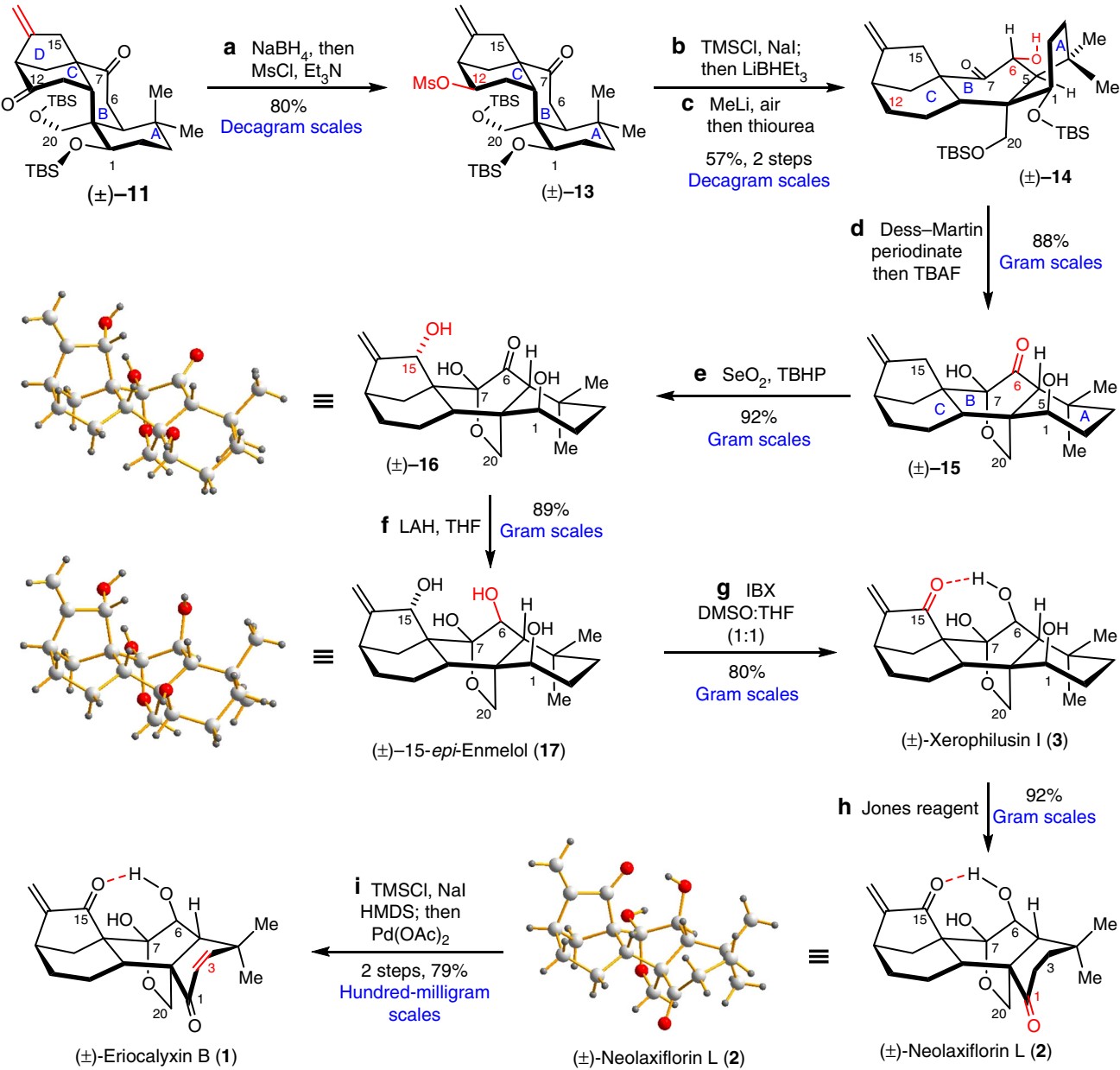

**Fig. 4** Total Syntheses of *Isodon* diterpenoids. With (±)−**11** as the common intermediate, the total syntheses of (±)-eriocalyxin B (**1**), (±)-neolaxiflorin L (**2**) and (±)-xerophilusin I (**3**) were achieved via a conformation and H-bond guided redox–relay strategy in 7–10 steps with 19–26% overall yields. The synthetic route can be operated in hundred-milligram-to-decagram scales

**Table 1 In vitro cancer growth inhibition of *Isodon* diterpenoids[a]**

| Compound | HL60 | SMMC7721 | A549 | MCF7 | SW480 |
|---|---|---|---|---|---|
| (−)−**1** | 2.989 | 2.573 | 4.843 | 1.133 | 0.598 |
| (±)−**1** | 4.281 | 3.627 | 8.075 | 1.956 | 0.981 |
| (±)−**2** | 5.817 | 7.586 | >15 | 5.774 | 2.939 |
| (±)−**3** | 5.549 | 7.712 | >15 | 4.131 | 2.380 |

[a] Human cancer cell lines were treated with different doses of natural (−)−**1** or synthetic (±)−**1**-**3** for 48 h. Cell viability was measured by MTT assay. Experiments were conducted in triplicate and results are expressed as mean IC$_{50}$ values in μM.

(Supplementary Figure 1), with the highest growth inhibitory effects in SW480 (Table 1). Natural (−)−**1** exhibited the highest growth inhibitory activity in all tested cancer cell lines, with IC$_{50}$

values as low as 0.598 μM. Synthetic (±)−**1** exhibited ~60–70% efficacy of (−)−**1**, and synthetic (±)−**2** and (±)−**3** exhibited similar levels of efficacy with moderate IC$_{50}$ values. These in vitro results are consistent with those reported in the literature[7,53,63]. In an additional experiment, we found that all four diterpenoids could significantly increase the expression of cleaved poly (ADP-ribose) polymerase (PARP) (Supplementary Figure 2), indicating potential involvement of apoptosis. Further flow cytometric investigation of Annexin V/Propidium Iodide (PI) stained cells showed that the diterpenoids could indeed induce apoptosis. Compared to control cells, treatment with (−)−**1** at 0.63 and 1.25 μM increased the proportion of apoptotic cells to 32.5% and 63.4%, respectively, while treatment with (±)−**1** increased apoptotic populations to 22.3% and 29.2%, respectively. Treatment with (±)−**2** at 2.5 and 5 μM resulted in apoptotic populations of 22.4% and 42.2%, respectively, while treatment with (±)

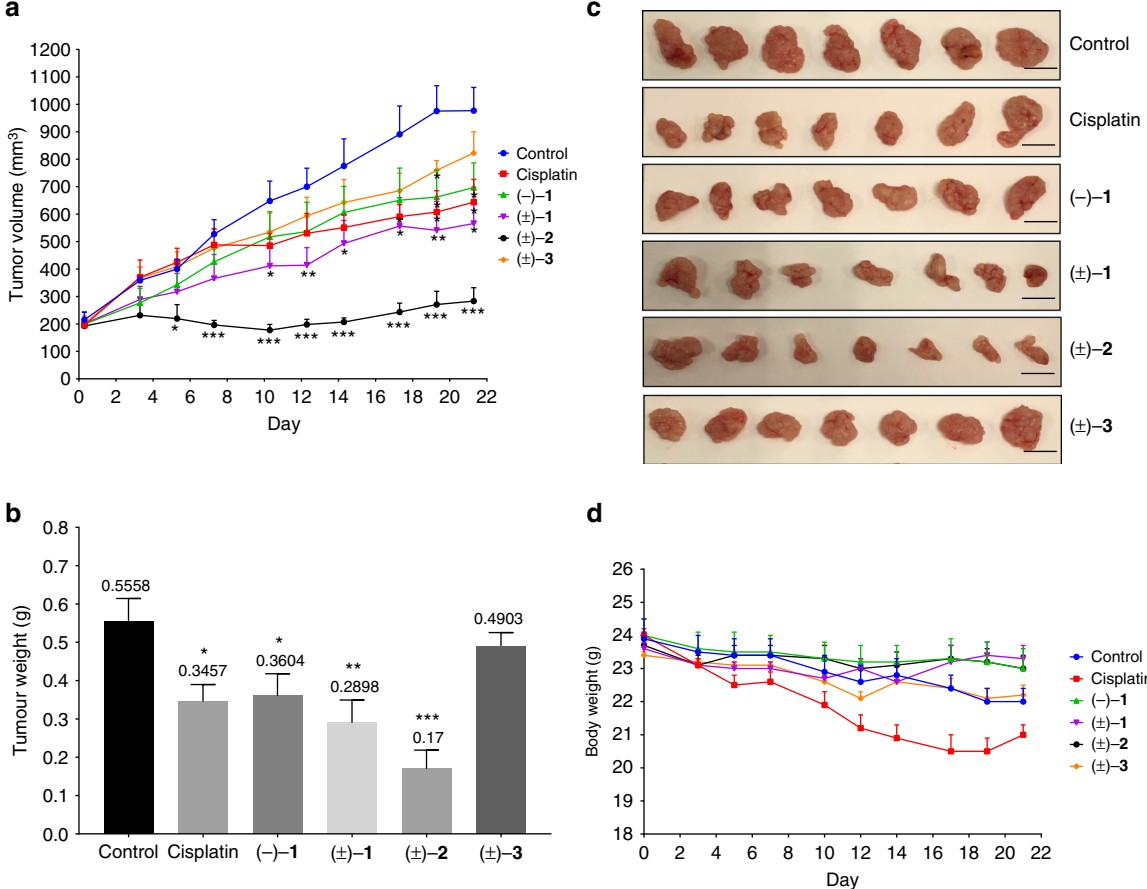

**Fig. 5** In vivo studies of *Isodon* diterpenoids. BALB/c nude mice xenografted with SW480 cells were randomised into different groups (*n* = 7 per group) and received intraperitoneal injections of compounds (natural (−)−**1** or synthetic (±)−**1**–**3**, 10 mg/kg) three times weekly. Mice injected with saline served as controls. The experimental period lasted for 21 days, after which the mice were killed and their tumours and organs harvested and weighed. Treatment with natural (−)−**1** and synthetic (±)−**1**–**3** could inhibit xenograft tumour growth, weight and volume, without significant effects on mouse body weight. **a** Tumour volumes throughout the 21-day experimental period. **b** Tumour weights at experimental endpoint. **c** Photographs of tumours at experimental endpoint. Scale bar, 10 mm. **d** Mouse body weight throughout the 21-day experimental period. For (**a**), (**b**) and (**d**), data are expressed as means from seven mice. Error bars represent ± SEM. *$P < 0.05$, **$P < 0.01$, ***$P < 0.001$ compared to control (Student's *t*-test)

−**3** resulted in apoptotic populations of 31.6% and 60.8%, respectively (Supplementary Figure 3). The above results show that the anti-cancer effects of the diterpenoids were potentially mediated via induction of apoptosis.

**In vivo studies**. To translate the above in vitro findings for in vivo biological relevance, we examined the inhibitory effects of the *Isodon* diterpenoids on SW480 tumour xenograft growth in nude mice. Surprisingly, (±)-neolaxiflorin L (**2**) exhibited remarkable efficacy in the animal study despite its moderate cell growth inhibitory activity. Average tumour volumes and weights were decreased by 71.0% and 69.4%, respectively ($P < 0.001$) when compared to control, an effect approximately twice the efficacy of cisplatin (34.1% and 37.8%, respectively) (Fig. 5A–C, Supplementary Table 1). Moreover, (±)−**2** exhibited no apparent toxicity in the mice, as no significant body or organ weight loss was observed (Fig. 5D and Supplementary Figure 4). These results indicate the importance of multilevel bio-evaluation in lead discovery from *Isodon* diterpenoids for the development of anti-cancer therapeutics.

**In vitro studies of synthetic (−)−1 and (+)−1**. It is also worth noting that the racemic eriocalyxin B (±)−**1** exhibited slightly higher efficacy than its natural form (−)−**1**. To quickly access

both enantiomers of eriocalyxin B in optically pure form, a considerable amount of (±)−**1** obtained by our synthetic route was separated using chiral HPLC column (Supplementary Figure 5) and the in vitro cancer inhibitory activity of (−)−**1** and (+)−**1** in SW480 were investigated (Supplementary Figure 6). Natural and synthetic (−)−**1** exhibited comparable levels of activity towards SW480 ($IC_{50}$ = 0.598 and 0.854 μM, respectively). Surprisingly, synthetic (+)−**1** also showed a significant growth inhibitory activity in SW480 ($IC_{50}$ = 1.670 μM). These results suggest that (−)−**1** and (+)−**1** may have different mechanisms of action.

## Discussion

A concise, versatile and scalable total synthesis of (±)-eriocalyxin B (**1**), (±)-neolaxiflorin L (**2**) and (±)-xerophilusin I (**3**) has been achieved in 14–17 steps with 3.9–5.3% overall yields from readily available substrates (±)−**4**, **5** and **7**. This scalable total synthesis enabled multilevel bio-evaluation of these low-abundance *Isodon* diterpenoids. In animal studies, (±)-neolaxiflorin L (**2**) exhibited remarkable in vivo anti-cancer efficacy with no apparent toxicity — properties which were not observed in cell line studies of this compound. These results demonstrate the importance of scalable synthesis and multilevel bio-evaluation in identification of lead compounds. Notably, in the study we have identified

neolaxiflorin L (**2**) as a promising anti-cancer drug candidate. We are currently preparing a focused library of natural and designed analogues of these natural products based on our synthetic platform for investigation of their biological targets and mode of actions. Moreover, optically pure synthetic (−)−**1** and (+)−**1** have been obtained by chiral HPLC and we found that the enantiomer of eriocalyxin B (+)−**1** also showed significant in vitro activity. Further study of the mechanisms of (−)−**1** and (+)−**1** is still ongoing in our laboratories.

## Methods

**General**. All air- and water-sensitive reactions were carried out under a nitrogen atmosphere with dry solvents under anhydrous conditions, unless otherwise noted. Reactions were monitored by thin-layer chromatography (TLC) carried out on 0.25 mm silica gel plates (60F-254) that were analysed by fluorescence upon 254 nm irradiation or by staining with $KMnO_4$ (200 mL $H_2O$ of 1.5 g of $KMnO_4$, 10 g of $K_2CO_3$ and 1.25 mL of 10% aqueous NaOH). Silica gel (60, particle size 0.0400.063 mm) was used for flash column chromatography. All the chemicals were purchased commercially and used without further purification. Anhydrous THF was distilled from sodium and benzophenone. Toluene was distilled over sodium. $CH_3CN$ and $CH_2Cl_2$ were distilled from calcium hydride. Molecular sieves were activated by heating at 200 °C for 12 h at ~1.0 torr. Yields refer to the isolated yields after silica gel flash column chromatography, unless otherwise stated. NMR spectra were recorded on either a 300 ($^1$H, 300 MHz; $^{13}$C, 76 MHz), 400 MHz ($^1$H, 400 MHz; $^{13}$C, 101 MHz) or 500 MHz ($^1$H, 500 MHz; $^{13}$C, 126 MHz) spectrometer. The correlation spectroscopy (COSY), heteronuclear single-quantum correlation spectroscopy (HSQC) and nuclear Overhauser effect spectroscopy (NOESY) were performed on a 400 or 500 MHz spectrometer. The following abbreviations were used to explain the multiplicities: s = singlet, d = doublet, t = triplet, q = quartet, m = multiplet, br = broad. High-resolution mass spectra were obtained from a matrix-assisted laser desorption/ionisation time-of-flight (MALDI-TOF) mass spectrometer. Crystallographic data were obtained from a single-crystal X-ray diffractometer. All the infrared (IR) spectra were recorded with a Fourier-transform infrared spectrometer. $^1$H and $^{13}$C NMR for all compounds, COSY, HSQC and NOSEY for compound (±)−**2**, (±)−**3**, (±)−**11**, (±)−**14**, (±)−**15** and (±)−**17** are provided, see Supplementary Figures 7–66. For Oak Ridge Thermal Ellipsoid Plot (ORTEP) of compounds (±)−**2**, (±)−**12**, (±)−**16** and (±)−**17**, see Supplementary Figures 67–70. For characterisation data of all compounds, see Supplementary Methods.

**Cell culture**. Five human cancer cell lines, HL60 (promyelocytic leukaemia), SMMC7721 (hepatocellular carcinoma), A549 (human alveolar basal epithelial adenocarcinoma), MCF7 (breast adenocarcinoma) and SW480 (colon adeno-carcinoma) were purchased from the American Type Culture Collection (Manassas, USA) and maintained in Dulbecco's Modified Eagle Medium (DMEM) or Roswell Park Memorial Institution (RPMI)-1640 medium (Life Technologies, USA) supplemented with 10% heat-inactivated Fetal Bovine Serum (FBS) and penicillin/ streptomycin (50 U/mL) at 37 °C, 5% $CO_2$. All cell lines were tested and confirmed to be free of mycoplasma contamination.

**Cell viability and cytotoxicity assays**. The cell viability of different cancer cell lines under drug treatment was determined using the MTT assay. $5 \times 10^4$ HL60 or $3–6 \times 10^3$ SMMC7721, A549, MCF7 or SW480 cells were seeded in each well of 96-well plates. After 24 h, cells were treated with different drugs at various concentrations for another 48 h. Cells were then treated with 0.5 mg/mL MTT at 37 °C for 4 h. Media was removed after incubation and DMSO was added to each well to dissolve the formazan crystals. Absorbance was measured at 570 nm using a Multiskan™ FC Microplate Photometer (ThermoFisher Scientific, USA). Three independent experiments were carried out and the $IC_{50}$ of the cell lines were calculated.

**Immunoblotting**. Cancer cell lines treated with various concentrations of natural (−)−**1** and synthetic (±)−**1**–**3** were harvested after 48 h. Cell pellets were lysed in Radioimmunoprecipitation assay (RIPA) buffer. DC Protein Assay (Bio-Rad, USA) was used to determine protein concentrations. Equal amounts of cell lysates were electrophoresed through Sodium Dodecyl Sulphate—Polyacrylamide gel electrophoresis (SDS-PAGE) gels and transferred onto polyvinylidene difluoride (PVDF) membranes (Bio-Rad, USA). The blots were then blocked in 5% non-fat skim milk and probed using cleaved-PARP, PARP (1:1000 dilutions; #7237, #9542 Cell Signalling Technology, USA) and β-actin (1:10000 dilution; sc-69879 Santa Cruz Biotechnology, USA) antibodies overnight. Blots were then incubated with the corresponding goat anti-rabbit (Santa Cruz Biotechnology, USA) or goat anti-mouse Horse radish peroxidase (HRP)-conjugated secondary antibodies (Life Technologies, USA). Protein bands were visualised using Clarity ECL Western blotting substrates (Bio-Rad, USA). Images were obtained using a ChemiDoc Imaging System (Bio-Rad, USA) and protein expression was analysed using Image

Lab software (Bio-Rad, USA). The unprocessed and uncropped scans original blots are shown in Supplementary Figures 71–75.

**Flow cytometric analysis of apoptosis**. SW480 cells were plated in 60 mm dishes at a density of $3 \times 10^5$ cells for 24 h. Cells were then treated with the diterpenoids at 0.63 and 1.25 μM ((−)−**1** and (±)−**1**) or 2.5 and 5 μM ((±)−**2** and (±)−**3**), for 48 h. At the end of the treatment period, the cells were washed with phosphate buffered saline (PBS), collected and stained with Dead Cell Apoptosis Kit with Annexin V-Alexa Fluor™ 488 & Propidium Iodide (Invitrogen) according to manufacturer's instructions. Flow cytometry was performed using an Accuri C6 Flow Cytometer (BD Biosciences), with detection of Annexin V-Alexa 488 fluorescence (FL1) and PI fluorescence (FL2). Data ($3 \times 10^4$ events per sample) were acquired using BD Accuri C6 software. A representative gating strategy for the flow cytometry experiments is shown in Supplementary Figure 76.

**Mouse xenograft tumour models**. BALB/c nude mice (male 8-week-old) were purchased from BioLASCO (Charles River, Taiwan). Mice were subcutaneously injected in their right flanks with $5 \times 10^6$ SW480 cells suspended in 100 μL serum free RPMI medium and mixed with an equal volume of Matrigel (Gibco, USA). After 14 days of inoculation, when tumours grew to an average volume of 180 $mm^3$, the mice were randomised into six different experimental groups ($n = 7$ mice per group). Natural (−)−**1** and synthetic (±)−**1**–**3** were prepared in ethanol and Cremophor EL (Sigma, USA) (1:1 ratio) and diluted to the desired concentration in 0.9% saline. Mice were injected intraperitoneally with 10 mg/kg of (−)−**1**, (±)−**1**–**3** or cisplatin three times weekly. A control group was injected with vehicle alone. Body weight and tumour volumes were measured three times weekly. Tumour volumes were calculated as (length × width$^2$)/2. After the 21 day experimental period, mice were killed, and their tumours and vital organs were harvested and weighed. Investigators were blind to the treatment groups during the experiments and data analysis. All animal experiments were approved by the Hong Kong Polytechnic University Animal Subjects Ethics Sub-committee and conducted in accordance with the Institutional Guidelines and Animal Ordinance of the Department of Health.

**HPLC**. A chiralpak IA column (0.46 cm I.D. × 15 cm L) was used and a racemic sample of (±)−**1** was eluted with 100% MeOH with flow rate equals 1.0 mL/min (detector = UV at 254 nm, temperature = 35 °C).

**Synthesis of compound (±)−4**. To a stirred solution of diisopropylamine (DIPA) (39.8 mL, 284 mmol) in THF (200 mL) was added n-BuLi (113.6 mL of a 2.5 M solution in hexanes, 284 mmol) at 0 °C. The solution was stirred at room temperature for 30 min and treated with ethyl acetate (26.9 mL, 284 mmol) slowly at −78 °C. After stirring for 15 min, acrolein (18.4 mL, 284 mmol) was added at −78 °C. The solution was stirred at −78 °C for 10 min, and the reaction was quenched by addition of a saturated $NH_4Cl$ aqueous solution (200 mL) at −78 °C. The mixture was warmed to room temperature and the aqueous phase was extracted with ethyl acetate (200 mL × 3). The combined organic extracts were washed with brine, dried over $MgSO_4$, filtered and concentrated. To a stirred solution of the residue in dichloromethane (200 mL) was added 4-methoxybenzyl 2,2,2-trichloroacetimidate (120 g, 426 mmol) and camphor sulfonic acid (CSA) (10 g, 42.6 mmol). The mixture was stirred at room temperature for 12 h and the reaction was quenched by addition of a saturated $NaHCO_3$ aqueous solution (200 mL). The aqueous phase was extracted with ethyl acetate (200 mL × 3). The combined organic extracts were washed with brine, dried over $MgSO_4$, filtered through a plug of silica gel and concentrated. To a stirred solution of the residue in THF (700 mL) was added LAH (9.7 g, 284 mmol) at 0 °C. After stirring at 0 °C for 4 h, the reaction was quenched by addition of ice and then a saturated sodium potassium tartrate aqueous solution (300 mL) at 0 °C. The aqueous phase was extracted with ethyl acetate (200 mL × 3). The combined organic extracts were washed with brine, dried over $MgSO_4$, filtered and concentrated. Silica gel flash column chromatography (hexanes/ethyl acetate = 3:1) of the residue gave a yellow oil (40 g, 180 mmol, 63%) as the product. (±)−**4**.

**Synthesis of compound 5**. To a stirred solution of dimethyl acetylmethylpho-sphonate (20.9 mL, 150 mmol) in THF (1.5 mL) was added NaI (15 g, 105 mmol) and DBU (22.4 mL, 100 mmol). The mixture was stirred at room temperature for 30 min, and then 3-bromopropyne (16 mL, 150 mmol) was added. The mixture was treated with additional NaI (15 g, 105 mmol), DBU (22.4 mL, 100 mmol) and 3-bromopropyne (16 mL, 150 mmol) every 12 h (×2). After the last addition of NaI/DBU/3-bromopropyne, the mixture was stirred at room temperature for 12 h before addition of acetic acid (10 mL). The mixture was filtered through a plug of diatomite and concentrated. Silica gel flash column chromatography (hexanes/ethyl acetate = 3:1) of the residue gave a yellow oil (24.3 g, 120 mmol, 80%) as the product.

**Synthesis of compound 7**. To a stirred solution of 2-(hydroxymethyl)-4,4-dimethyl cyclohex-2-en-1-one (33 g, 214 mmol) in DMSO (60 mL) was added IBX (76 g, 273 mmol) slowly at 0 °C. After TLC analysis showed the consumption of the alcohol intermediate, the mixture was diluted with diethyl ether (100 mL) and

water (100 mL). The aqueous phase was extracted with diethyl ether (200 mL × 3). The combined organic extracts were washed with brine (200 mL × 3), dried over MgSO₄, filtered and concentrated. The residue was filtered through a plug of silica gel and concentrated to provide a colourless oil (29.3 g, 193 mmol, 90%) as the product.

**Synthesis of compound (±)−6**. To a stirred solution of PCC (42.5 g, 180 mmol) in dichloromethane (250 mL) was added a solution of (±)−4 (20 g, 90 mmol) in dichloromethane (300 mL) dropwise at 0 °C. The mixture was stirred at room temperature until TLC analysis showed consumption of the starting material. Then the mixture was filtered through a plug of silica gel and concentrated. To a stirred solution of 5 (18.2 g, 89 mmol) in THF (1 L) was added NaI (13.4 g, 89 mmol) and DBU (12.1 mL, 89 mmol) at room temperature. After stirring for 30 min (formation of a white precipitation), a solution of the above residue (16.5 g, 80.9 mmol) in THF (50 mL) was added and the resulting mixture was stirred at room temperature for 10 h. The reaction was then quenched by addition of a saturated NH₄Cl aqueous solution (500 mL), and extracted with ethyl acetate (200 mL × 3). The combined organic extracts were concentrated. To a stirred solution of the residue in THF (400 mL) was added a 1 N solution of aqueous HCl (400 mL) at room temperature. After stirring for 12 h, the reaction was quenched by addition of a saturated NaHCO₃ aqueous solution and the aqueous phase was extracted with ethyl acetate (200 mL × 3). The combined organic extracts were washed with brine, dried over anhydrous MgSO₄, filtered and concentrated. Silica gel flash column chromatography (hexanes/ethyl acetate = 10:1) of the residue gave a yellow oil (18.1 g, 60.7 mmol, 67%) as the product (*E*-isomer only).

**Synthesis of compound (±)−9**. To a stirred solution of (±)−6 (18.1 g, 60.7 mmol) and triethylamine (83 mL, 607 mmol) in dichloromethane (500 mL) was added triisopropylsilyl trifluoromethanesulfonate (TIPSOTf) (27.8 mL, 91.1 mmol) slowly at 0 °C. After stirring at room temperature for 30 min, the mixture was concentrated. The residue was dissolved in hexanes (500 mL) and filtered through a plug of silica gel and concentrated. To a stirred solution of the residue in toluene (500 mL) was added 7 (11.0 g, 72.8 mmol). The mixture was heated under reflux for 12 h, then cooled to room temperature and concentrated. The residue was filtered through a plug of silica gel and gave a pale yellow oil (36.8 g, 60.7 mmol) as the product ((±)−8, a 5:1 mixture of *exo/endo* isomers). To a stirred solution of (±)−8 in THF (400 mL) and H₂O (100 mL) was added NaBH₄ (2.3 g, 60.7 mmol) slowly at 0 °C. After stirring at room temperature for 3 h, the reaction was quenched by addition of brine (200 mL), and the aqueous phase was extracted with ethyl acetate (200 mL × 3). The combined organic extracts were washed with brine, dried over MgSO₄, filtered, and concentrated. Silica gel flash column chromatography (hexanes/ethyl acetate = 3:1) of the residue gave the product (16.0 g, 26.2 mmol, 58% for 2 steps) as a 1:1 mixture of diastereomers. The two diastereomers of (±)−9 can be used for the next step. For the sake of characterisation, the two diastereomers were separated by silica gel column chromatography.

**Synthesis of compound (±)−10**. To a stirred solution of (±)−9 (26.3 g, 43.1 mmol) and triethylamine (295 mL, 860 mmol) in dichloromethane (300 mL) was added *t*-butyldimethyl trifluoromethanesulfonate (TBSOTf) (29.9 mL, 129.1 mmol) slowly at 0 °C. After stirring at room temperature for 3 h, the reaction was quenched by addition of water (300 mL) and the aqueous phase was extracted with dichloromethane (200 mL × 3). The combined organic extracts were washed with brine, dried over MgSO₄, filtered and concentrated. To a stirred solution of the residue in dichloromethane (700 mL) and H₂O (171 mL) was added DDQ (25.9 g, 129 mmol) slowly at room temperature. The reaction was stirred at room temperature until TLC showed completion of the reaction. The reaction was then quenched by addition of a saturated NaHCO₃ aqueous solution and the aqueous phase was extracted with hexanes (200 mL × 3). The combined organic extracts were washed with brine, dried over MgSO₄, filtered and concentrated. Silica gel flash column chromatography (hexanes/ethyl acetate = 30:1) of the residue gave a yellow oil (24.7 g, 34.5 mmol, 80% for 2 steps) as the product.

**Synthesis of compound (±)−11**. To a stirred solution of LiBr (4.39 g, 50.2 mmol) and Me₂AlCl (100.6 mL, 100.6 mmol) in dichloromethane (300 mL) was added a solution of (±)−10 (18 g, 25.1 mmol) in dichloromethane (200 mL) dropwise at room temperature over 45 min. After addition, the solution was stirred at room temperature for 20 min, and then poured into a mixture of hexanes (300 mL), dichloromethane (200 mL) and saturated sodium potassium tartrate aqueous solution (100 mL) at 0 °C. After stirring at room temperature for 8 h, the aqueous phase was extracted with ethyl acetate (200 mL × 3). The combined organic extracts were washed with brine, dried over MgSO₄, filtered and concentrated. Silica gel flash column chromatography (hexanes/ethyl acetate = 25:1) of the residue gave a white solid (9.0 g, 16.0 mmol, 65%) as the product.

**Synthesis of compound (±)−12**. To a stirred solution of (±)−11 (175 mg, 0.3 mmol) in dichloromethane (10 mL) was added *p*-toluenesulfonic acid (TsOH) (100 mg, 0.6 mmol) at room temperature. The mixture was stirred at room temperature until TLC analysis showed consumption of the starting material. The solution was then concentrated, and silica gel flash column chromatography

(hexanes/ethyl acetate = 10:1) of the residue gave a yellow oil (130 mg, 0.29 mmol, 98%) as the product.

**Synthesis of compound (±)−13**. To a stirred solution of (±)−11 (14.3 g, 25.5 mmol) in dichloromethane (150 mL) and methanol (150 mL) was added NaBH₄ (0.96 g, 25.5 mmol) slowly at 0 °C. After stirring at room temperature for 10 h, the reaction was quenched by addition of brine (100 mL). The aqueous phase was extracted with ethyl acetate (200 mL × 3), and the combined organic extracts were washed with brine, dried over MgSO₄, filtered and concentrated. To a stirred solution of the residue and triethylamine (59.1 mL, 255 mmol) in dichloromethane (250 mL) was added methylsulfonyl chloride (MsCl) (5.8 g, 51 mmol) slowly at 0 °C. After stirring at room temperature for 1 h, the reaction was quenched by addition of a saturated NH₄Cl aqueous solution (200 mL), and the aqueous phase was extracted with ethyl acetate (200 mL × 3). The combined organic extracts were washed with brine, dried over MgSO₄, filtered and concentrated. Recrystallisation of the residue with hexanes gave a white solid (13.0 g, 20.3 mmol, 80%) as the product. (±)−13.

**Synthesis of compound (±)−14**. To a stirred solution of (±)−13 (15.7 g, 24.4 mmol) in CH₃CN (35 mL) was added NaI (18.2 g, 122 mmol), HMDS (30.5 mL, 146 mmol) and trimethyl chloride (TMSCl) (18.3 mL, 146 mmol) at 0 °C. The mixture was warmed to room temperature and stirred for 10 h. The reaction was then diluted with hexanes (150 mL) and quenched by addition of H₂O (30 mL). The organic phase was washed with water (30 mL × 3 or until it became clear). The combined aqueous solution was extracted with hexanes (150 mL × 2) and the combined organic extracts were washed with brine, dried over MgSO₄, filtered and concentrated. To a stirred solution of the residue in hexanes (155 mL) was added LiBHEt₃ (244 mL of a 1 M solution in THF, 244 mmol) at room temperature. The mixture was stirred at 55 °C for 8 h, and then exposed to air without stirring at room temperature overnight. The reaction was quenched by addition of water (70 mL), and the aqueous phase was extracted with hexanes (150 mL × 3). The combined organic extracts were washed with brine, dried over MgSO₄, filtered and concentrated. Silica gel flash column chromatography (hexanes/ethyl acetate = 80:1) of the residue gave a colourless oil (10.5 g, 17.1 mmol, 70%) as the product ((±)−18, for the structure, ¹H and ¹³C NMR of this compound, see Supplementary Figures 30 and 31). To a stirred solution of (±)−18 (12.0 g, 19.5 mmol) in THF (200 mL) was added MeLi (19.5 mL of a 2.5 M solution in THF, 48.8 mmol) at room temperature. After stirring for 2 h, air was bubbled into the solution until it turned brown. Then the mixture was treated with thiourea (5.9 g, 78.1 mmol) and stirred for 10 min at room temperature. Addition of hexanes (450 mL) leads to a white precipitation. The white suspension was filtered through a plug of silica gel and concentrated. Silica gel flash column chromatography (hexanes/ethyl acetate = 80:1) of the residue gave a yellow oil (8.96 g, 16.0 mmol, 82%) as the product.

**Synthesis of compound (±)−15**. To a stirred solution of (±)−14 (5.13 g, 9.12 mmol) in dichloromethane (90 mL) was added Dess-Martin periodinane (5.9 g, 12.8 mmol) at 0 °C. After stirring at room temperature for 30 min, the mixture was filtered through a plug of silica gel and concentrated. To a stirred solution of the residue in THF (20 mL) was added TBAF (25.7 mL, 25.7 mmol) at room temperature. The solution was concentrated after stirring for 48 h. Silica gel flash column chromatography (hexanes/THF = 3:1) of the residue gave a white solid (2.7 g, 8.0 mmol, 88%) as the product.

**Synthesis of compound (±)−16**. To a stirred solution of (±)−15 (2.4 g, 7.5 mmol) in dichloromethane (75 mL) was added *t*-butyl peroxide (3.4 mL, 18.75 mmol) and SeO₂ (164 mg, 1.5 mmol) at room temperature. The mixture was stirred for 24 h (a white precipitation was resulted). After removal of the volatiles, the resulting solid was washed with hexane/ethyl acetate = 3:1 (15 mL × 4) to provide a white solid (2.4 g, 6.8 mmol, 92%) as the product.

**Synthesis of (±)−15-*epi*-enmelol (17)**. To a stirred solution of (±)−16 (1.5 g, 4.5 mmol) in THF (900 mL) was added LAH hydride (0.57 g, 14.9 mmol) slowly at room temperature. After stirring for 1.5 h, the reaction was quenched by addition of a 4 N HCl aqueous solution (10 mL) at 0 °C. The aqueous solution was extracted with ethyl acetate/THF = 2:1 (200 mL × 4). The combined organic extracts were washed with brine, dried over MgSO₄, filtered and concentrated. Silica gel flash column chromatography (hexanes/THF = 1:1) of the residue gave a white solid product (1.4 g, 4.0 mmol, 89%) as the product.

**Synthesis of (±)−xerophilusin I (3)**. To a stirred solution of (±)−17 (2.4 g, 6.84 mmol) in DMSO (300 mL) and THF (300 mL) was added IBX (3.84 g, 13.7 mmol) slowly at 0 °C. The mixture was stirred at room temperature until TLC showed consumption of the starting material. The mixture was diluted with ethyl acetate (600 mL) and reaction was quenched by addition of a saturated NaHSO₃ aqueous solution (20 mL). The aqueous phase was extracted with THF/ethyl acetate = 1:1 (500 mL × 3). The combined organic extracts were washed with brine (500 mL × 5), and then concentrated. Silica gel flash column chromatography (hexanes/THF = 1:1) of the residue gave a white solid (1.9 g, 5.48 mmol, 80%) as the product.

**Synthesis of (±)−neolaxiflorin L (2)**. To a stirred solution of (±)−3 (1.0 g, 2.47 mmol) in acetone (250 mL) was added Jones reagent (1.7 mL of a 2.9 M in acetone/water solution, 4.9 mmol) at 0 °C. After stirring at 0 °C for 15 min, the reaction was quenched by addition of a saturated NaHCO₃ aqueous solution (2.5 mL) and iso-propanol (3.5 mL). After dilution with hexanes (500 mL), the solution was filtered through a plug of silica gel and then concentrated. Silica gel flash column chromatography (hexanes/ethyl acetate = 5:1) of the residue gave a white solid (0.82 g, 2.28 mmol, 92%) as the product.

**Synthesis of (±)-eriocalyxin B (1)**. To a stirred mixture of (±)−2 (0.27 g, 0.78 mmol), NaI (1.05 g, 7.02 mmol) and HMDS (2.45 mL, 11.7 mmol) in CH₃CN (4 mL) was added TMSCl (0.53 mL, 3.9 mmol) slowly at 0 °C. After stirring at room temperature for 12 h, the reaction was diluted with hexanes (30 mL), and quenched by addition of water (5 mL). The aqueous phase was extracted with hexanes (20 mL × 3). The combined organic extracts were washed with water until clear, and then washed with brine, dried over MgSO₄, filtered and concentrated. To a stirred solution of the residue in CH₃CN (8 mL) was added Pd(OAc)₂ (525 mg, 2.34 mmol) at room temperature. After stirring at room temperature for 6 h, the reaction was quenched by addition of a 0.5 N HCl aqueous solution (7 mL). The aqueous phase was extracted with ethyl acetate (20 mL × 3). The combined organic extracts were washed with brine, dried over MgSO₄, filtered and concentrated. Silica gel flash column chromatography (hexanes/ethyl acetate = 2:1) of the residue gave a white solid (0.21 g, 0.62 mmol, 79%) as the product.

**Data availability**. The X-ray crystallographic coordinates for structures reported in this article have been deposited at the Cambridge Crystallographic Data Centre (CCDC), under deposition number CCDC 1556435 for (±)-neolaxiflorin L (2), CCDC 1556434 for (±)−12, CCDC 1556436 for (±)−16 and CCDC 1556433 for (±)-15-epi-enmelol (17). These data can be obtained free of charge from The CCDC via www.ccdc.cam.ac.uk/data_request/cif. The authors declare that other data supporting the findings of this study are available within the paper and its supplementary information files and also are available from the corresponding author upon request.

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

## Acknowledgements

This paper is dedicated to Professor Craig J. Forsyth on the occasion of his 60th birthday. This work is financially supported by the National Natural Science Foundation of China (21272012, 21502002 and 21572006), Natural Science Foundation of Guangdong Province, China (2017A030313042), Shenzhen Science, Technology and Innovation Committee (JCYJ2015062611042525, JCYJ20160330095659560 and JCYJ20160428153421635), Chinese Postdoctoral Science Foundation (2015M580910), Hong Kong Research Grants Council (Collaborative Research Grants, CRF PolyU11/ CRF/13E; General Research Fund, GRF PolyU 12103515), Hong Kong Polytechnic University [University Research Facility for Chemical and Environmental Analysis (UCEA), Area of Excellence Grants (1-ZVGG)] and Peking University Shenzhen Graduate School. A special acknowledgement is made to Dr. Wesley Ting-kwok Chan (Hong Kong Polytechnic University) for the X-ray crystallography of compounds (±)−**2**, (±)−**12** and (±)−**16**–**17**.

## Author contribution

C.S.L. and W.C.S.T. wrote the manuscript and compiled the Supplementary Information. All authors contributed to reading and editing of the manuscript and Supplementary Information. L.Z., W.M. and M.Z. conducted the chemical reactions and prepared the synthesis section of the Supplementary Information. W.M. optimised the DA cycloaddition and the Mukaiyama–Michael/carbocyclization cascade cyclization in the core synthesis. L.Z. optimised the conformation and hydrogen-bond-guided redox–relay sequence and finished the syntheses of (±)−**1**–**3**. M.Z. optimised the scale-up synthesis and prepared required compounds for multilevel biological assays. M.M.L.L., W.Y.W. and B.D.C. conducted the cell-based assays and animal studies, and prepared the biological section of the Supplementary Information. Q.Y. and W.T.W. solved the X-ray structures and prepared the X-ray section of the Supplementary Information.

## Additional information

**Competing interests:** The authors declare no competing interests.

