## [Peer Review File · Nature Communications]

Reviewers' comments:

Reviewer #1 (Remarks to the Author):

In this manuscript, Zhu et al. report the synthesis of a series of agents based on natural products from plant genus *Isodon*. While this work describes considerable organic synthesis efforts, the below comments are focused only on the biological testing aspects.

The most promising agents were tested *in vitro* using the MTT assay, a well-established and reliable assay of mitochondrial activity. However, this assay does not by itself distinguish between cytotoxic and cytostatic effects. Because results are calculated relative to the vehicle-treated control (which continues to grow during the course of the experiment), results are better described as growth inhibition rather than cytotoxicity, as no parallel assessments of actual cell death were conducted. The number of times the assay was separately performed for each cell line also needs to be included. The *in vivo* testing is compelling, but again is done only once, with a relatively small cohort size particularly for a xenograft model. That effects were discernable with only 7 mice per group is promising, but a repeat experiment using larger numbers would contribute greatly to the impact of these experiments. On a related note, actual p-values should be presented rather than upper limits.

The point that compound (2) was notably more effective *in vivo*, in contrast to *in vitro* results, is intriguing and underscores how accepted methods of preclinical evaluation can be misleading. The authors' point that molecules should be tested in multiple different assays is well-taken. How does cisplatin perform under these same conditions? Is its clinical efficacy predicted by the cell lines selected?

The PARP immunoblots add little, as the great majority of cytotoxic agents will cause an increase in PARP cleavage; this is an expected downstream effect as cells die, and also predictable based on MTT data. More work is needed to make any convincing statement regarding mechanism of action.

Overall, the assays selected are very reasonable, albeit limited, first steps. Recognizing that this is mainly a report describing organic synthesis, greater interest and impact of this research would be realized if some additional information could be added regarding any of the following: hypothesized mechanism of action (different cell/molecular effects across the cell lines tested?); therapeutic index (relative effects on cancer vs. normal cells?); pharmacokinetic parameters (how were the dose and schedule for *in vivo* studies selected?); longer term efficacy and toxicity (what happens if mice that do not meet euthanasia criteria continue past 21 days, either with or without continued treatment?); bioavailability (can cell or organ uptake explain the discordance between the *in vitro* and *in vivo* results?) and toxicity (studies in normal, immune-competent mice that include impact on hematopoietic compartment as well as main organ sites).

This is a nicely written paper that clearly represents a large amount of work, but without any of these questions addressed, it will remain of limited interest to most readers of *Nature Communications* and perhaps deserves a more specialized journal.

Reviewer #2 (Remarks to the Author):

Zhu and coworkers report on the development of a scalable strategy that has allowed for the synthesis of several related polycyclic diterpene natural products isolated from the *Isodon* species. This species of plant is a prodigious producer of highly oxygenated terpenes, many of which possess potent and potentially useful biological properties. In fact, the plants have been used in traditional medicine for centuries. The compounds thus represent important possible leads for the

development of new, modern therapeutics. A realization of this potential has been made difficult due to a) only small quantities are available for natural sources, and b) the structures of the compounds are very complicated rendering fully synthetic approaches very challenging. Because of the latter challenges, many of these compounds have become attractive targets for efforts in total synthesis dating back several decades. In more recent times, this family has drawn much attention and has led to many total syntheses of various family members. Much of the recent efforts have been highlighted in some recent reviews (none of which were cited here, oddly).

Despite the great strides made in recent years, there still remains much room for improvement in the development of efficient routes that are scalable and also generalizable. The current submission seeks to do just this, and to a large degree the work is very successful to this end. The authors report the preparation of three related diterpene natural products, (\pm)-ericalyxin B, (\pm)-neolaxiflorin L and (\pm)-xerophilusin in between "13-15" steps. The difference between these three structures are, however, trivial oxidation state adjustments. Essentially, a concise approach to one particular core has been devised so in this sense it remains to be seen if the strategy can be generalized to other family members with more intricate skeletons. Having said that, the current effort is very effective and utilizes some creative bond disconnections. A particular highlight is the rapid rise in complexity that is obtained by the 5-step sequence beginning with a convergent Diels-Alder reaction and ending with a quite remarkable LA-catalyzed Mukaiyama-Michael/alkylation cascade that produces a complex tetracyclic scaffold in a scalable and efficient manner (i.e., compound 11). While the configuration of C5 is incorrect at this point, epimerization is achieved later during oxidation at C6 (14 to 15). The final stages of the syntheses involve some interesting chemoselective oxidations. The lack of oxidation of the C6 alcohol within 3 upon exposure to Jones reagent is pretty surprising.

With material in hand, the authors conduct a series of biological evaluations both in vitro and in vivo. The initial cell studies mostly recapitulated known values in the literature, but this is important since in some cases this is not seen (i.e., with maoecrystal V).

Some comments:

Earlier in my review, I say "13-15" steps because by my counting this is not factual; there is an all too frequent trend in the organic synthesis community to count several steps as one step just because the intermediates were not purified. If a solvent is changed and/or an aqueous workup took place then that is one step. Thus, an actual real step count here is closer to 20 steps (to compound 17), 21 steps (to 3) and 23 (to 1) – and this steps count does not include the synthesis of the "starting material" 4 (which takes 3 steps to make and is in the SI). I very much dislike the trend among organic synthetic chemists to undercount their steps for the sake of trying to make a route look more impressive; it's disingenuous and does a disservice to the field.

Why use "decagram" as a scale designation? The first step is done on 20 grams, that should be sufficient. 20 grams uses fewer pixels and has fewer syllables than 2 decagrams.

In Figure 4, some of the structures have no numbers (the last few compounds, including the natural products also lack names). The authors need to fix this.

Overall, this is a good piece of work. The synthesis contains a nice sequence of reactions that efficiently construct the core scaffold in an effective manner. The cascade reaction featuring the Mukaiyama-Michael/alkyne alkylation is very effective, allowing a rather rapid rise in complexity. From a strategy perspective, these disconnections are quite interesting. Despite these observations, however, the work lacks a certain degree of novelty in that it doesn't utilize or develop any truly novel transformations. Preparation of the molecules as racemates also diminishes some of the excitement around the approach, even though there are benefits to having both enantiomers at once... The biological activity reported is promising and highlights the usefulness in conducting tissue-based assays if possible. Overall, despite some reservations this is a nice piece of synthetic chemistry that delivers on its stated goal to provide material for further biological investigations. On this basis it is of sufficient quality for acceptance in Nature

Communications following some of the modifications suggested above.

Reviewer #3 (Remarks to the Author):

Prof. Lee and coworkers utilized the intermolecular Diels-Alder cycloaddition and the intramolecular cascade Mukaiyama Michael/carbocyclization as key steps to furnish the scalable total syntheses of (±)-ericalyxin B, (±)-neolaxiflorin L and (±)-xerophilusin I. The manuscript is carefully prepared, and well organized.

However, due to the key synthetic strategy of this work has been disclosed by the authors in 2013 (Zhu, L. Z., Han, Y. J., Du, G. Y., Lee, C. S. *Org. Lett.*, 15, 524-527 (2013)), and the asymmetric total syntheses have not been achieved. In addition, the authors used the racemic form of natural product to evaluate the biological activity, which may not reflect the real biological activity of the natural product. I therefore do not recommend this manuscript in the current form for publications in *Nature Communication*, but after a major revision to address the following issues it could be reconsidered.

- 1) Asymmetric total synthesis needs to be achieved;
- 2) Using the synthesized and optical pure natural product to do the biological testing;
- 3) The following references should be included in the text of page 2, line 66, "The intriguing structure and biological activity of *Isodon* diterpenoids has attracted considerable efforts towards their synthesis.¹³⁻⁴⁶
a) He, C., Hu, J. L., Wu, Y. B., Ding, H. F. *J. Am. Chem. Soc.* 139, 6098–6101 (2017). b) He, C., Bai, Z. B., Hu, J. L., Wang, B. N., Xie, H. J., Yu, L., Ding, H. F. *Chem. Commun.*, 53, 8435--8438 (2017).
- 4) Page 5, figure 3, change "H2O" to "H₂O".
- 5) Page 5, figure 3, according to the procedure in the SI, the transformation of compound 4 to compound 6 took three steps, sincere the authors worked up the reaction for each step. Please make revision accordingly;
- 6) Page 6 line 166, page 7 figure 4, the authors took 2 steps to finish the total synthesis of compound 1 from compound 2 instead of only one step.
- 7) Page 6 line 170, the sentence of "without using any protecting group" should be removed. Sincere the authors utilized the TBS protecting group to control the regioselectivity in the DMP oxidation.
- 8) Page 20, please check the reference 47.

Response to Reviewer 1:

1. The most promising agents were tested in vitro using the MTT assay, a well-established and reliable assay of mitochondrial activity. However, this assay does not by itself distinguish between cytotoxic and cytostatic effects. Because results are calculated relative to the vehicle-treated control (which continues to grow during the course of the experiment), results are better described as growth inhibition rather than cytotoxicity, as no parallel assessments of actual cell death were conducted. The number of times the assay was separately performed for each cell line also needs to be included.

Response:

Thank you, we appreciate and agree with your concerns that our MTT results only show the growth inhibitory effects of the diterpenoids and not their cytotoxicity. Because of this, we have conducted Annexin V/PI flow cytometric analysis to investigate the cytotoxic activity of the diterpenoids. SW480 cells were selected for this part of the experiment, as the diterpenoids showed the highest efficacy in this cell line in MTT experiments, and it was also the cell line used in tumor xenograft experiments. The results of the flow cytometric analysis for apoptosis clearly demonstrate that compared to control cells, treatment with (-)-1 at 0.63 and 1.25 μM increased the proportion of apoptotic cells to 32.5% and 63.4% respectively, while treatment with (\pm)-1 increased apoptotic populations to 22.3% and 29.2% respectively. Treatment with (\pm)-2 at 2.5 and 5 μM resulted in apoptotic populations of 22.4% and 42.2%

respectively, while treatment with (\pm)-**3** resulted in apoptotic populations of 31.6% and 60.8% respectively (Figure S3). These results have been incorporated into the main manuscript and Supplementary materials, Figure S3.

For the MTT assays, three independent experiments were conducted, and this information has been included in the Methods section of the manuscript.

2. The in vivo testing is compelling, but again is done only once, with a relatively small cohort size particularly for a xenograft model. That effects were discernable with only 7 mice per group is promising, but a repeat experiment using larger numbers would contribute greatly to the impact of these experiments.

Response:

*Thank you for your comments. Before conducting the animal studies, we calculated the required number of animals per group to ensure we could obtain a statistically significant study. Using G*Power software with an effect size of $f=0.65$, α of 0.05, power level of 80% and group number of 6, it was calculated that an n number of 7 mice per group would be sufficient to reach statistical significance. From the results of our animal studies, we have clearly demonstrated the reduction of tumour volumes and weights after treatment with (-)-**1** and (\pm)-**1-3**. Because of the above results, we believe that we have provided strong evidence support the in vivo anti-cancer effects of (-)-**1** and (\pm)-**1-3**.*

However, we do agree that it would be a good idea to perform additional studies with larger numbers of animals, and in the second phase of our ongoing studies, we plan to conduct these additional studies.

3. On a related note, actual p-values should be presented rather than upper limits.

Response:

We have taken into account your suggestion and have edited Table S1 to include the actual p-values for tumor volumes and weights at the experimental endpoint.

4. The point that compound (2) was notably more effective in vivo, in contrast to in vitro results, is intriguing and underscores how accepted methods of preclinical evaluation can be misleading. The authors' point that molecules should be tested in multiple different

assays is well-taken. How does cisplatin perform under these same conditions? Is its clinical efficacy predicted by the cell lines selected?

Response:

Thank you for your comments. In our paper, we were not trying to suggest that in vitro results could be misleading, instead, we would like to address the importance of in vivo studies in drug discovery and development. For example, in our study, the in vitro efficacy of (±)-2 did not completely reflect its in vivo activity. That this might also be the case for a certain small percentage of compounds, underlines the importance of multilevel bio-evaluation.

*According to our literature review, cisplatin exhibited an IC₅₀ of 7.7 μM after 48 hours treatment in SW480 cells ¹ (conditions matching our in vitro experiments), and has an effective dosage in xenograft mouse models of 5-10mg/kg ²⁻³. Although the IC₅₀ of cisplatin in SW480 is higher than the diterpenoids **1**, **2** and **3**, in in vivo studies, cisplatin exhibited similar or better tumor inhibitory activity than diterpenoids **1** and **3**. Again, this underlines the importance of in vivo studies for drug discovery and development.*

- (1) Alonso-Castro, A. J., Dom íguez, F., & Garc í-Carranc á A. Rutin exerts antitumor effects on nude mice bearing SW480 tumor. Archives of medical research, **44**, 346-351 (2013).*
- (2) He, G., He, G., Zhou, R., Pi, Z., Zhu, T., Jiang, L., & Xie, Y. Enhancement of cisplatin-induced colon cancer cells apoptosis by shikonin, a natural inducer of ROS in vitro and in vivo. Biochemical and biophysical research communications, **469**, 1075-1082 (2016).*
- (3) Coxon, A., Ziegler, B., Kaufman, S., Xu, M., Wang, H., Weishuhn, D., & Polverino, A. Antitumor activity of motesanib alone and in combination with cisplatin or docetaxel in multiple human non-small-cell lung cancer xenograft models. Molecular cancer, **11**, 70. (2012).*

5. The PARP immunoblots add little, as the great majority of cytotoxic agents will cause an increase in PARP cleavage; this is an expected downstream effect as cells die, and also predictable based on MTT data. More work is needed to make any convincing statement regarding mechanism of action.

Response:

We agree with your comments that if a mechanism of action is to be convincingly argued, more work should be included. To address this, we have conducted additional Annexin V/PI flow cytometry experiments to further investigate and confirm the mechanism of apoptosis in

the diterpenoid-mediated cancer cell death. SW480 cells were selected for this part of the experiment, as the diterpenoids showed the highest efficacy in this cell line in MTT experiments, and it was also the cell line used in tumor xenograft experiments. The results of the flow cytometric analysis for apoptosis clearly demonstrate that compared to control cells, treatment with (–)-1 at 0.63 and 1.25 μM increased the proportion of apoptotic cells to 32.5% and 63.4% respectively, while treatment with (±)-1 increased apoptotic populations to 22.3% and 29.2% respectively. Treatment with (±)-2 at 2.5 and 5 μM resulted in apoptotic populations of 22.4% and 42.2% respectively, while treatment with (±)-3 resulted in apoptotic populations of 31.6% and 60.8% respectively (Figure S3). These results have been incorporated into the main manuscript and Supplementary materials, Figure S3.

6. Overall, the assays selected are very reasonable, albeit limited, first steps. Recognizing that this is mainly a report describing organic synthesis, greater interest and impact of this research would be realized if some additional information could be added regarding any of the following:

- a. hypothesized mechanism of action (different cell/molecular effects across the cell lines tested?);

Response:

Thank you for your comments and suggestions. The focus of this paper is to demonstrate scalable synthesis can enable multi-level biological evaluations of natural products, which could speed up the process of discovery of lead compounds. Regarding the mechanism of action of the diterpenoids 1, 2 and 3, this is a part of ongoing studies that are currently in progress.

- b. therapeutic index (relative effects on cancer vs. normal cells?);

Response:

Thank you for your suggestion. In our animal studies in the tumor xenograft mouse model, treatment with cisplatin led to a 4.6% reduction of body weight compared to control mice, while treatment with (–)-1, (±)-1, (±)-2, and (±)-3, led to body weight increases of 4.6%, 6.1%, 1.2% and 1.2% respectively (Figure 5D). Organ weights were also unaffected by treatment with the diterpenoids (Figure S4). The above results suggest that (–)-1 and (±)-1–3 have lower toxicity to animals than cisplatin and they may not display significant adverse side effects on the normal cells of the animals.

- c. pharmacokinetic parameters (how were the dose and schedule for in vivo studies selected?);

Response:

For our initial animal studies, we selected a dose and treatment schedule (10mg/kg, 3 injections/week) that matched that of our positive control, cisplatin, which would allow us to compare the anti-cancer effect of the diterpenoids with the commonly used anti-cancer therapeutic.

In general, a cisplatin dose of 10mg/kg has previously been used in xenograft mouse studies¹.

- (1) *He, G., He, G., Zhou, R., Pi, Z., Zhu, T., Jiang, L., & Xie, Y. Enhancement of cisplatin-induced colon cancer cells apoptosis by shikonin, a natural inducer of ROS in vitro and in vivo. Biochemical and biophysical research communications, 469, 1075-1082 (2016).*

- d. longer term efficacy and toxicity (what happens if mice that do not meet euthanasia criteria continue past 21 days, either with or without continued treatment?);

Response:

Thank you for your suggestion. As the tumour size of control mice was nearing the ethical limit of 1000mm³, we could not continue the study any longer, and stopped at 21 days. In general, 21 -28 days is standard protocol for xenograft studies. In order to address the reviewer's comments on the long-term toxicity and efficacy of the compounds, in the second phase of our studies, we plan to perform animal experiments that could extend to at least 28 days.

- e. bioavailability (can cell or organ uptake explain the discordance between the in vitro and in vivo results?)

Response:

Thank you for your comments. In terms of the bioavailability, we agree with your comments on their importance, and will work on the pharmacokinetics studies for preclinical development of the diterpenoids 1, 2 and 3 in the second phase of our studies.

- f. toxicity (studies in normal, immune-competent mice that include impact on hematopoietic compartment as well as main organ sites).

Response:

Thank you for your comments. We agree that toxicity is one of the important parameters that should be assessed for preclinical drug development, and we will work on these studies for the preclinical development of the diterpenoids 1, 2 and 3 in the second phase of our studies.

The aim of this paper was to demonstrate scalable synthesis can enable multi-level biological evaluations of natural products, which could speed up the process of discovery of lead compounds. In the second phase of our studies, which is currently ongoing, we will focus on the preclinical development of the diterpenoids 1, 2 and 3. As suggested by the reviewer, investigations into the toxicity, pharmacokinetics and detailed mechanisms will be performed in the second phase, with the aim of developing the compounds as anti-cancer therapeutics.

Response to Reviewer 2:

1. Much of the recent efforts have been highlighted in some recent reviews (none of which were cited here, oddly).

Response:

Thank you for your comment. We have added the recent reviews on the synthetic efforts on ent-kaurene related natural products in references 13-16.

2. The difference between these three structures are, however, trivial oxidation state adjustments. Essentially, a concise approach to one particular core has been devised so in this sense it remains to be seen if the strategy can be generalized to other family members with more intricate skeletons.

Response:

*Thank you for your comment. The major focus on this paper is to develop a concise synthetic approach towards the 7,20-epoxy-ent-kauranoids and to demonstrate that our synthetic approach can provide a stable supply for (±)-ericalyxin B (1), (±)-neolaxiflorin L (2) and (±)-xerophilusin I (3) for multilevel biological evaluations. Although these three structures are similar with different oxidation states in the A ring, selective oxidation of the C1 hydroxyl of 3 in the presence of the free hydroxyl at C6 is not a trivial task. Moreover, we have also mentioned in the text about the extremely low isolation yields of neolaxiflorin L (2) and xerophilusin I (3) (only 3 and 12 mg respectively from 10 kg of dried leaves of *I. ericalyx* var. *laxiflora*). The materials that we obtained from our synthesis allowed multilevel biological evaluations of these natural products and, indeed, we found that neolaxiflorin L (2) showed the best in vivo efficacy with no apparent*

toxicity in the mice.

3. Earlier in my review, I say “13-15” steps because by my counting this is not factual; there is an all too frequent trend in the organic synthesis community to count several steps as one step just because the intermediates were not purified. If a solvent is changed and/or an aqueous workup took place then that is one step. Thus, an actual real step count here is closer to 20 steps (to compound 17), 21 steps (to 3) and 23 (to 1) – and this steps count does not include the synthesis of the “starting material” 4 (which takes 3 steps to make and is in the SI). I very much dislike the trend among organic synthetic chemists to undercount their steps for the sake of trying to make a route look more impressive; it’s disingenuous and does a disservice to the field.

Response:

Thank you for your comments and suggestions. Compound (±)-4, 5 and 7 are readily available compounds that have been commonly used as starting materials in total synthesis of natural products. We did state clearly that our synthesis required “11-13” steps from these starting materials in the text. We have also added the compound numbers for the “readily available substrates” in the conclusion for clarification. Generally, transformations that involved unstable intermediates, such as aldehydes and silyl enol ethers (which did not required a formal workup), will not be considered as a step. The comments from Reviewer 3 also indicated that this method of step count is quite acceptable in the scientific committee.

For the steps count, we also agree with the comments from Reviewer 3. It required 2 steps from transforming compound 2 to 1. We have made the corrections in the text and figure. However, we think that there should be only 2 steps for the transformations from compound 4 to 6 since we only did a quick filtration of the aldehyde intermediate without workup. Therefore, the total number of steps for our synthesis has been corrected to 14 steps for (±)-xerophilusin I (3), 15 steps for (±)-neolaxiflorin L (2) and 17 steps for (±)-ericalyxin B (1) from compound (±)-4, 5 and 7. These corrections have been made in the text and figures.

4. Why use “decagram” as a scale designation? The first step is done on 20 grams, that should be sufficient. 20 grams uses fewer pixels and has fewer syllables than 2 decagrams.

Response:

Thank you for your comments. We used the term “decagram scales” due to the fact that we have carried out the transformation in different scales. For example, we have done the first step in 20, 30 and 50 grams. The 20 gram scale in the SI is the most frequently scale that we have been done.

5. In Figure 4, some of the structures have no numbers (the last few compounds, including the natural products also lack names). The authors need to fix this.

Response:

Thank you for your comment. We have added the compound number and names of the natural products in Figure 4.

6. Despite these observations, however, the work lacks a certain degree of novelty in that it doesn't utilize or develop any truly novel transformations. Preparation of the molecules as racemates also diminishes some of the excitement around the approach, even though there are benefits to having both enantiomers at once...

Response:

Thanks you for your comments. Our synthesis featuring a concise seven-step sequence for the core synthesis as well as the H-bond and conformation guided redox relay strategy for installing the correct oxidation at C1, C6 and C15. Moreover, one of the key transformations in our synthetic strategy is the Mukaiyama Michael/carbocyclization. This cascade cyclization was anticipated the requirement of a dual-mode Lewis acid, which could induce cyclization by forming the σ -complex (Mukaiyama Michael addition) and π -complex (carbocyclization). To our surprise, a strong σ -Lewis acid Me_2AlCl can induce both Mukaiyama Michael and carbocyclization cyclization with LiBr as an additive. There is only one report in the literature for using Me_2AlCl in carbocyclization (ref 57), which gave a 6-endo product, contrasting the 5-exo product in our case. A detail mechanistic study on the mechanism of this key transformation is ongoing in our laboratory. Moreover, we have added a brief discussion including comparison with our previous study and the literature to the text.

*An asymmetric synthesis could be readily achieved by an asymmetric Diels-Alder cyclization between compound **6** and **7** with a chiral catalytic system, such as Cu(II) /bis-oxaoline or Yb(III) /pyBox. Theoretically, both enantiomers of the natural products could be obtained by using both enantiomers of the ligands. However, since the results of the animal model showed that the racemic mixture of the natural products exhibited a slightly higher efficacy, we are keen on quickly obtain both enantiomers of the natural products for biological assays. The quickest way to obtain the two enantiomers in optically pure form is to scale up the racemic synthesis and separated the two enantiomers by chiral HPLC. Indeed, we have successfully obtained optically pure form of both enantiomer of **1**, and found that synthetic (-)-**1** exhibited comparable in vitro anti-cancer activity with natural (-)-**1** and (+)-**1** also showed significant in vitro anti-cancer activity. These results indicated that the two enantiomers may have different mechanism of actions. We are currently scaling up the synthesis and obtained a considerable amount of both enantiomers of **1** by*

chiral HPLC for animal study.

Response to Reviewer 3:

1. However, due to the key synthetic strategy of this work has been disclosed by the authors in 2013 (Zhu, L. Z., Han, Y. J., Du, G. Y., Lee, C. S. *Org. Lett.*, 15, 524-527 (2013)), and the asymmetric total syntheses have not been achieved. In addition, the authors used the racemic form of natural product to evaluate the biological activity, which may not reflect the real biological activity of the natural product.

Response:

*Thank you for your comments and suggestions. Our previous work in 2013 (Zhu, L. Z., Han, Y. J., Du, G. Y., Lee, C. S. *Org. Lett.*, 15, 524-527 (2013)) involved a Diels-Alder/carbocyclization cascade cyclization strategy. The key transformation of this work is actually a Mukaiyama Michael/carbocyclization. There are significant differences between these two cascade cyclization reactions. Indeed, we found that our previous strategy, Diels-Alder/carbocyclization cascade cyclization, failed to construct the CD ring system due to the fact that the C9-C11 carbon-carbon bond cannot be established by intermolecular reaction, such as Diels-Alder cycloaddition. Therefore, we decided to construct the C9-C11 carbon-carbon bond in the early stage and switch to the intramolecular Mukaiyama Michael addition for constructing the C ring. Unfortunately, the intramolecular Mukaiyama Michael addition required a strong σ -Lewis acid (Me_2AlCl), which is not favor for the 5-exo carbocyclization. Surprisingly, we found that the cascade cyclization went effectively with $\text{Me}_2\text{AlCl}/\text{LiBr}$, which is unprecedented in the literature. A detail mechanistic study of this transformation is ongoing in our laboratory. Moreover, we have added a brief discussion including comparison with our previous study and the literature to the text.*

*As we response to the comment of Reviewer 2, an asymmetric synthesis could be readily achieved by an asymmetric Diels-Alder cyclization between compound **6** and **7** with a chiral catalytic system, such as Cu(II) /bis-oxaoline or Yb(III) /pyBox. Since we are keen quickly prepare both enantiomers of the natural products for biological assays, we have decided to obtain the two enantiomers in optically pure form by separation of the racemic mixture by chiral HPLC. Indeed, we have successfully obtained optically pure form of both enantiomer of **1**, and found that synthetic (-)-**1** exhibited comparable in vitro anti-cancer activity with natural (-)-**1** and (+)-**1** also showed significant in vitro anti-cancer activity. These results indicated that the two enantiomers may have different mechanism of actions. We are currently scaling up the synthesis and obtained a considerable amount of both enantiomers of **1** by chiral HPLC for animal study.*

2. Asymmetric total synthesis needs to be achieved;

Response:

Thank you for your suggestion. As we mentioned above, we are keen on the studying the mechanism of the two enantiomers of the natural products. Instead of developing an asymmetric synthesis, we think that the quickest way to obtain the two enantiomers in optically pure form is to scale up on the racemic synthesis and separated the two enantiomers using chiral HPLC. This strategy also matches the main theme of this manuscript.

3. Using the synthesized and optical pure natural product to do the biological testing;

Response:

*Thank you for your suggestion. As we mentioned above, we have successfully obtained optically pure (-)-**1** and (+)-**1** for a comparison study with the natural (-)-**1**. The in vitro study of synthetic (-)-**1** and (+)-**1** have been reported in the text of the manuscript and the SI.*

4. The following references should be included in the text of page 2, line 66, “The intriguing structure and biological activity of Isodon diterpenoids has attracted considerable efforts towards their synthesis.13-46

a) He, C., Hu, J. L., Wu, Y. B., Ding, H. F. J. Am. Chem. Soc. 139, 6098–6101 (2017). b) He, C., Bai, Z. B., Hu, J. L., Wang, B. N., Xie, H. J., Yu, L., Ding, H. F. Chem. Commun., 53, 8435--8438 (2017).

Response:

Thank you for your suggestion. The two references have been added.

5. Page 5, figure 3, change “H2O” to “H₂O”.

Response:

Thank you for your suggestion. The typo has been fixed.

6. Page 5, figure 3, according to the procedure in the SI, the transformation of compound 4 to compound 6 took three steps, sincere the authors worked up the reaction for each step. Please make revision accordingly;

Response:

Thank you for your comment and suggestion. For the transformation from compound 4 to 6, the aldehyde intermediate did a quick filtration without workup. We have changed the steps to 2 in the

text and figure for this transformation.

7. Page 6 line 166, page 7 figure 4, the authors took 2 steps to finish the total synthesis of compound 1 from compound 2 instead of only one step.

Response:

Thank you for your suggestion. We agreed with the comment and have changed the number of steps to 2 for the transformation of compound 2 to 1.

8. Page 6 line 170, the sentence of “without using any protecting group” should be removed. Sincere the authors utilized the TBS protecting group to control the regioselectivity in the DMP oxidation.

Response:

*Thank you for your suggestion. Actually, we did not introduce any additional protecting group in the conformation and H-bond guided redox relay starting from compound (\pm)-**II**. We also agreed that the TBS group already in compound (\pm)-**II** has effect of the selective oxidation. So we decided to delete the phrase “without using any protecting group” in the text.*

9. Page 20, please check the reference 47.

Response:

Thank you for your suggestions. The above changes have been changed according to your suggestions.

Thank you once again for allowing us to revise and polish our manuscript. We look forward to your favorable consideration for our manuscript.

Yours sincerely,
Chi-Sing Lee

Reviewers' Comments:

Reviewer #1:

Remarks to the Author:

The authors have responded reasonably to each of my comments, but have not added much new data. Due to the normal variability in animal studies, I remain unconvinced that a single xenograft study with 7 mice per group is sufficient to draw firm conclusions, regardless of what statistical software suggests. However, their responses are useful and should definitely be incorporated into the final version if the editor decides to move forward with publication.

Reviewer #2:

Remarks to the Author:

In their revised submission, the authors have taken some steps to address the comments I made in my initial review.

1. The addition of the citations to recent reviews was made.
2. The issue regarding step-count was not resolved, as the authors believe that if an intermediate is not purified then the reaction that made the compound does not count as a step. Such liberal step-count analyses are now commonplace in the literature, much to the chagrin of this reviewer for our field. I still believe that such step-counting does a disservice to synthetic efforts, but will not make my argument further.
3. The use of decagram as a term, while somewhat silly in the opinion of this reviewer, was justified.
4. The authors fixed compound numbering issues.

Overall, the work is of high quality and the syntheses are very efficient (even if they are longer in steps than the authors believe), allowing some initial biological evaluations. The synthetic work contains some nice reactions sequences; A particular highlight is the rapid rise in complexity that is obtained by the 5-step sequence beginning with a convergent Diels-Alder reaction and ending with a quite remarkable LA-catalyzed Mukaiyama-Michael/alkylation cascade that produces a complex tetracyclic scaffold in a scalable and efficient manner (i.e., compound 11). As stated in my initial review, the work is a nice example of strategy design. Despite the issues of step-count, which chemists nowadays think is subjective, I am not opposed to acceptance of the work in Nature Communications

Reviewer #3:

Remarks to the Author:

The revised version has addressed my previous questions, and the current manuscript is acceptable for publishing on Nature Commun.

REVIEWERS' COMMENTS:

Reviewer #1 (Remarks to the Author):

The authors have responded reasonably to each of my comments, but have not added much new data. Due to the normal variability in animal studies, I remain unconvinced that a single xenograft study with 7 mice per group is sufficient to draw firm conclusions, regardless of what statistical software suggests. However, their responses are useful and should definitely be incorporated into the final version if the editor decides to move forward with publication.

Thank you for your comments. We regret that you remain skeptical about the animal study group size, however, as was mentioned, before conducting the studies; we calculated the required number of animals per group to ensure that a statistically significant study could be achieved.

We appreciate your initial comments and agree that the follow up responses are indeed beneficial to the paper. As per your suggestion, these additions have been incorporated into the final text.

Reviewer #2 (Remarks to the Author):

In their revised submission, the authors have taken some steps to address the comments I made in my initial review.

1. The addition of the citations to recent reviews was made.
2. The issue regarding step-count was not resolved, as the authors believe that if an intermediate is

not purified then the reaction that made the compound does not count as a step. Such liberal step-count analyses are now commonplace in the literature, much to the chagrin of this reviewer for our field. I still believe that such step-counting does a disservice to synthetic efforts, but will not make my argument further.

3. The use of decagram as a term, while somewhat silly in the opinion of this reviewer, was justified.

4. The authors fixed compound numbering issues.

Overall, the work is of high quality and the syntheses are very efficient (even if they are longer in steps than the authors believe), allowing some initial biological evaluations. The synthetic work contains some nice reactions sequences; A particular highlight is the rapid rise in complexity that is obtained by the 5-step sequence beginning with a convergent Diels-Alder reaction and ending with a quite remarkable LA-catalyzed Mukaiyama-Michael/alkylation cascade that produces a complex tetracyclic scaffold in a scalable and efficient manner (i.e., compound 11). As stated in my initial review, the work is a nice example of strategy design. Despite the issues of step-count, which chemists nowadays think is subjective, I am not opposed to acceptance of the work in Nature Communications

Thank you for your comments and being supportive for publication of our work. We fully agree with you that we should be more objective for the step counts. Unfortunately, there is no general guideline for this issue. We can only follow the guidelines that commonly used in the scientific community.

Reviewer #3 (Remarks to the Author):

The revised version has addressed my previous questions, and the current manuscript is acceptable for publishing on Nature Commun.

Thank you for your comments and being supportive for publication of our work.

Thank you once again for your comments and being supportive to our work.

Yours sincerely,

Chi-Sing Lee